# Brain Inspired Cortical Coding Method for Fast Clustering and Codebook Generation

**DOI:** 10.3390/e24111678

**Published:** 2022-11-17

**Authors:** Meric Yucel, Serdar Bagis, Ahmet Sertbas, Mehmet Sarikaya, Burak Berk Ustundag

**Affiliations:** 1National Software Certification Research Center, Istanbul Technical University, Istanbul 34469, Turkey; 2Computer Engineering Department, Istanbul University-Cerrahpasa, Istanbul 34320, Turkey; 3Computer Engineering Department, Istanbul Technical University, Istanbul 34469, Turkey; 4Materials Science & Engineering Department, University of Washington, Seattle, WA 98195, USA

**Keywords:** clustering, codebook generation, cortical coding model, distortion, entropy maximization, execution time, feature extraction, machine learning, real-time execution, vector quantization

## Abstract

**Highlights:**

**What are the main findings?**
A cortical coding method is developed inspired by the network formation in the brain, where the information entropy is maximized while dissipated energy is minimized.The execution time in the cortical coding model is far superior, seconds versus minutes or hours, compared to those of the frequently used current algorithms, while retaining comparable distortion rate.

**What is the implication of the main finding?**
The significant attribute of the methodology is its generalization performance, i.e., learning rather than memorizing data.Although only vector quantization property was demonstrated herein, the cortical coding algorithm has a real potential in a wide variety of real-time machine learning implementations such as , temporal clustering, data compression, audio and video encoding, anomaly detection, and recognition, to list a few.

**Abstract:**

A major archetype of artificial intelligence is developing algorithms facilitating temporal efficiency and accuracy while boosting the generalization performance. Even with the latest developments in machine learning, a key limitation has been the inefficient feature extraction from the initial data, which is essential in performance optimization. Here, we introduce a feature extraction method inspired by energy–entropy relations of sensory cortical networks in the brain. Dubbed the brain-inspired cortex, the algorithm provides convergence to orthogonal features from streaming signals with superior computational efficiency while processing data in a compressed form. We demonstrate the performance of the new algorithm using artificially created complex data by comparing it with the commonly used traditional clustering algorithms, such as Birch, GMM, and K-means. While the data processing time is significantly reduced—seconds versus hours—encoding distortions remain essentially the same in the new algorithm, providing a basis for better generalization. Although we show herein the superior performance of the cortical coding model in clustering and vector quantization, it also provides potent implementation opportunities for machine learning fundamental components, such as reasoning, anomaly detection and classification in large scope applications, e.g., finance, cybersecurity, and healthcare.

## 1. Introduction

Feature extraction plays a key role in developing highly efficient machine learning algorithms, where time complexity constitutes the major part of the energy consumption thereby limiting the widespread implementation, especially in real-time applications. Identifying the key features derived from the input data would normally yield dimensionality reduction without losing significant information which, in turn, provides less effort, increases the speed of learning, and enables the generalization of the model. Among the various types of feature extraction approaches, vector quantization and clustering are widely used for compression, data mining, speech coding, face recognition, computer vision, and informatics [1,2,3,4]. Vector quantization is a process of mapping a set of data that forms a probabilistic distribution of input vectors by dividing them into a smaller set that represents the input space according to some proximity metric [1,5,6]. During data encoding, the vectors representing the input data are then compared to the entries of a codebook containing representative vectors, where the ones similar to the signal are transmitted to the receiver. The transmitted index is decoded on an identical codebook at the receiver side, thus an approximation to the original signal is reconstructed. This process requires fewer bits than transmitting the raw data in the vector and, hence, leads to the increased efficiency in data compression [7,8]. The key to effective data encoding and compression, therefore, is a good codebook generation. Moreover, one of the most crucial problems in clustering is to identify the number of clusters in advance. Having such a priori knowledge would increase the performance of most clustering algorithms. In recent years, due to this challenge, automatic clustering algorithms, which are designed to identify optimal cluster size in advance, gained significant interest [9]. The method we herein present is a dynamic, self-organized, online clustering method that does not take cluster size in advance, and is suitable for automatic clustering.

There are various algorithms developed to improve codebook performance. Among these, K-means clustering is a partition-based clustering algorithm which uses a combinatorial optimization approach. Despite having low time complexity, the commonly used model K-means has a local minimum problem; therefore, to reach the global minimum, the algorithm has to be rerun several times [10]. The performance is strictly defined by the initial codebook [11]. An example of hierarchical clustering algorithms is Birch, balanced iterative reducing and clustering hierarchy, with improved performance when used with PNN, Pairwise Nearest Neighbour. Gaussian Mixture Model, GMM, is a model-based clustering model [12]. The details of these algorithms are described in Section 2.5). The overarching goal of all vector quantization models is to generate a good codebook with high efficiency (low bitrate), high signal quality (low distortion), and low time and space complexity (computation, memory, and time delay) [1,13]. These algorithms and their adapted variations developed during the last two decades are rather simple to use. There are, however, trade-offs among these factors in each of the models that facilitate their use in specific implementations. GMM and PNN, for example, usually perform better than K-means by generating good codebooks, but the computational complexities of GMM and PNN are rather high and, therefore, they are challenging to use for large datasets. The time complexity of the algorithms, in particular, exponentially increases with the size of the input data and the dimensions of the vectors. Birch has a very low time complexity and a single read of input data is enough for clustering; additional passes of input data may improve the clustering performance. However, Birch is order-sensitive and mostly clustering performance decreases when the clusters are not spherical [14]. In short, although each algorithm offers specific advantages for a given application, their common limitation is related to a combination of key factors including generalization performance and execution time.

In this study, cortical coding (cortex) based codebook generation model is inspired from morphological adaptation in the sensory cortex, and it models simplified behavior of neuronal interactions instead of their cellular model. Following the definition of the information entropy by Shannon, the coding methods in communication and their performance measure use entropy as a metric depending on the probabilistic features of the signal patterns [15]. In the biological cortical networks, the maximum thermodynamic entropy also tends to yield maximum information entropy, improving energy efficiency via the effective use of the structure in-memory operational capabilities, improving the capacity. The cortical coding model we introduce here is for codebook generation, clustering and feature extraction that mimics the thermodynamic and morphologically adaptive features of the brain driven by the received signals. To demonstrate the effectiveness of the cortical coding model, we evaluate its clustering and generalization performance compared to the common codebook-based clustering algorithms. The brain-inspired cortical coding model mimics several biological principles in the codebook generation, including:(a)Neurons minimize dissipated energy in time by increasing the most common features of the probabilistic distribution of the received signal pattern properties [16];(b)Neurons develop connections along the electric field gradient mostly selected by spine directions [17];(c)A signal may be received by multiple neurons through their dendritic inputs while absorbing the relevant features in a parallel manner that specifies their selectivity [18];(d)New connections between axon terminals and the dendritic inputs of the neurons are made for higher transferability of the electric charge until maximum entropy, or equal thermal dissipation, is reached within the cortex [19];(e)New neurons may develop in the brain from stem cells when there is a lack of connectivity in certain preferred directions [20,21];(f)The overall entropy tends to increase by developing and adapting the cortical networks with respect to the received signal within a given environment [22].

Incorporating these principles, in the simplest model, each cortex operates with minimum energy encoding the maximum possible information content. The basis of high efficiency and fast analysis of transfer, storage, and execution of information is attributed to the self-organized, hierarchically structured, and morphogenetic nature of the neural networks in the brain. A general representation of the neural network which forms a cortex in a biological brain (e.g., rats) is given in Figure 1a, which shows that a neuron consists of soma, the neural node, with outgoing axon that carries the information and distributes through neural branches each of which may form a synapse at the ends of spines emanating from dendrites which carries the filtered information into the next soma which, in turn, acts as a new node. In this model, therefore, each neural node, soma or neural body, controls the information traffic connecting to other nodes through axons while the information is received through the dendrites and their extension, the spines (Figure 1a). Here, the most suitable distribution for modeling a given set of data is the one with the highest entropy among all those that satisfy the constraints of the acquired prior knowledge. Neuronal connections may form by synaptic terminals among the spines based on the new signals that become trigger zones summing up the collective inhibitory or excitatory signals incoming to a soma [19,22]. If the total accumulated charge level in a certain time window of the signal exceeds the threshold, the neuron will fire an action potential down the axon. In this system, new signal pattern would lead to the formation of a new set of neuronal connections which tend to find the most relevant neural bodies, thereby forming a hierarchical network, each constituting a related set of information.

Inspired by biology, we developed a codebook generation algorithm for feature extraction that uses wavelet transformation for frequency and localization analyses. The discrete wavelet packet transform (DWPT) is preferred because it can incorporate more detailed frequency analysis of the input data. The generalized flow of the brain-inspired cortical coding network formation from input data and feature extraction to cortical coding network training and neural network construction forming a cortex is schematically shown in Figure 1b (also, see Section 2.2 in Methods). In short, as seen in the input layers of the sensory cortices, the codebook is structured as a hierarchical n-array tree with each node holding a relevant wavelet coefficient [16]. The key advantage here is that the model is dynamic and self-organized without requiring a look-ahead buffer, which otherwise would need to incorporate all the input dataset before clustering them, a major impediment for real-time operation. In the brain-inspired cortical coding network, the framed data are sampled from an input signal where the frame size is determined by a specific vector. The normalized data are transformed with DWPT, and the wavelet coefficients of each frame are used for cortical coding network training, hierarchically starting with the lowest ending with the highest frequency components, i.e., from the roots to the leaves. As summarised in Figure 1c, the brain-inspired cortical coding network created in this work, therefore, is hierarchical, sorted with n-array type tree structure which has both spine and cortex nodes where the former is a candidate to be the latter once matured. The spine nodes and the sibling cortex nodes are sorted according to their node value. The tree structure is self-established, having coefficients from the lowest to the highest frequency from the roots to the leaves connected by the hierarchically formed branches, to each of which, when training is complete, an index number is assigned and each forms a specific code-word for the codebook. The cortical coding network training, therefore, is dynamic, self-organized, and an adaptive framework formed through a dissipated energy minimization activity. In this model, the information entropy is maximized, which may be dubbed as a thermomorphic process. Below, we discuss the construction of the cortical coding algorithm followed by its comparative implementation to demonstrate its efficiency in distortion rate and time complexity. Founded on these key factors in encoding, constituting the most energy-consuming step in machine learning, cortical coding model opens up new opportunities in practical real-time implementations in the future.

## 2. Methods

### 2.1. Datasets Used in Testing the Algorithms

Two different types of datasets were used for experiments, as described below.

#### 2.1.1. Datasets with Basic Periodic Waves Form

The first type of data used was a set of synthetic signals that consist of basic waves, i.e., sinus, sawtooth, square. The generated dataset was periodic and deterministic, therefore, is suitable for proof of concept studies. The signals were randomly generated with varying periods and amplitudes, e.g., ten different amplitudes varying from 1000 to 20,000, four different periods, 6, 8, 10, and 12, where the vector size was 8. This number was chosen since it is complex enough to compare algorithms and not too complex to have excessively long execution times for the comparison of the algorithms. The generated signal was sampled with 8 kHz taking into consideration that the frequency range used in the tests covers peak frequencies in human speech. It will, therefore, be easier to implement the cortical coding algorithm for coding speech or voice in the future. The occurrence of the signals is equally probable. In total there were 120 different signals (see Figure 2a). Not only were the generated signals used in tests individually, but also all shifts of signals were considered. For instance, if the generated signal had a sinus wave followed by a square wave, since each signal had eight data samples, there was a total of 16 samples. The window size was eight and the windows were shifted one all along the generated signals resulting in nine different signals ([0..8], [1..9], [2,10], …, [8,15]). Therefore, taking into account all shifts and varying periods and amplitudes resulted in a large number of unique input signals.

#### 2.1.2. Generation of the Lorenz Chaotic Dataset

The Lorenz system is a three-dimensional simplified mathematical model for studying complex phenomena, originally developed for atmospheric convection but also applied to electrical circuits, chemical reactions, and chaotic systems [23,24]. The model consists of three ordinary differential equations, the so-called Lorenz equations, and describes the rate of change of three quantities with respect to time. The Lorenz system is nonlinear, deterministic and non-periodic. Chaotic data never repeats itself and, therefore, it generates suitable temporal data for experiments and it is commonly used in many research fields. As an alternative to the digital data generated based on simple sinusoidal waveforms (Figure 2a), here we also introduced a one-dimensional chaotic waveform, *x* series Lorenz, which represents complex data, and we examined the performance of each of the four algorithmic platforms quantitatively in comparison to the cortical coding method. The one-dimensional data would represent voice, music, or text, although more complex cases could also be implemented using the same Lorenz equations, for example in video streaming or other complex phenomena where the dimensionality would increase. In the original Lorenz equations written below,
(1)$$\begin{array}{cc} \frac{\partial x}{\partial t} & =\sigma (y-x) \end{array}$$
(2)$$\begin{array}{cc} \frac{\partial y}{\partial t} & =x(\rho -z)-y \end{array}$$
(3)$$\begin{array}{cc} \frac{\partial z}{\partial t} & =xy-\beta z, \end{array}$$
the parameters *x*, *y*, and *z* are interdependent, and the σ, ρ, and β are constants, whose values are chosen such that the system exhibits chaotic behavior. In our data generation, the standard values are used, σ, ρ, and β are 10, 28, and 8/3, generating the complex wave behavior represented in the example in Figure 2b. The initial conditions of *x*, *y*, and *z* are taken from the known values [25]. The signal amplitude is multiplied by 10,000 to make the data amplitude values to be within a comparable range of the basic wave. Each clustering algorithm used the same dataset and two parameters, execution time and distortion rate, were compared, as shown in Figure 9 and are discussed in the main text.

### 2.2. Components of Cortical Coding Network

The cortical coding network has a hierarchical n-array tree structure. There are two different types of nodes in the network: cortex and spine nodes. Cortex nodes are the main nodes of the cortical coding tree, and spine nodes are those that are candidates to become cortex nodes. A cortex node has one parent node and may have multiple progeny cortex and spine nodes. Cortex node has three main variables which are node value, effective covering range, and pass count. Node value is a scalar value representing one element of the incoming vector data. For instance, if the node is a kth level node of the cortical coding tree, then the value of the node represents the kth coefficient of the incoming wavelet data. Both the cortex and spine nodes have the effective range parameter. The effective dynamic covering range parameters set the boundaries on which the node is active. Range parameter is limited within two values: $r_{init}$ and $r_{limit}$. $r_{init}$ is the initial value of the range and it is the upper bound for the range. The node’s active range is initially maximized when the node is generated. While training the node, the range parameter narrows down to the minimum boundary of the range, $r_{limit}$. Narrowing down a covering range is limited in order to prevent the generation of unlimited number of nodes. The pass count parameter is used for counting how many times a node is triggered and updated. The adaptation of the node (changing the node’s value according to the incoming data) is controlled with the pass count parameter. If the node is well trained, which means that pass count is high, then the adaption to new coming data within its range is slow. Correspondingly, if a node is not well trained, the adaptation is fast. Both cortex and spine nodes have pass count parameters.

Spine nodes, different to cortex nodes, have a maturity parameter which controls the training level of the node. If a spine node is well trained (if maturity passes a certain threshold), then it evolves into a cortex node. Unlike cortex nodes, spine nodes do not have progeny nodes and when the training of the network is finished, spine nodes are deleted. The nodes of the cortical coding tree are presented in Figure 3.

### 2.3. Cortical Coding Network Training Algorithm—Formation of Spines and Generation of the Network

The developed method is a combination of transformation and codebook coding. Here, the input signal was framed by predetermined window size. Then, it was normalized depending on the input signal type and compression ratio. We used DWPT with Haar kernel that was applied to the normalized signal towards achieving detailed frequency decomposition [26,27]. The wavelet coefficients were applied in a hierarchical order to the cortical coding tree. From root to leaves, the cortical coding tree holds wavelet coefficient(s) hierarchically ordered from low to high-frequency components. While the root node does not have any coefficient, the first level nodes have the lowest frequency coefficients, whereas last level nodes (leaves) have the highest frequency coefficients. In the beginning, the cortical coding tree has only the root node and is fed by the incoming data, with a wide range of wavelet coefficients. The nodes develop depending on the input data in a fully dynamic, self-organized, and adaptive way aiming to maximize the information entropy by minimizing the dissipation energy. Here, each cortex node has spine nodes that are generated according to the incoming wavelet coefficients and each spine node is a candidate to be a cortex node. If a spine node is triggered and well-trained, then it can turn into a cortex node. The significance of our approach is that only frequent and highly trained data can generate a node rather than every piece of data creating a node. The importance of this is that the cortical coding tree becomes highly robust eliminating the noise.

The evolution of a spine node to a cortex node is controlled with a first-order Infinite Impulse Response (IIR) Filter. IIR Filters have infinite impulse response and the output of the filter depends on both previous outputs and inputs [28]. The maturity of the spine node is calculated with the simple two-tap IIR filter given in Equation (Equation 4), where y[n] is the maturity level, x[n] is the dissipated energy and $k_{l}$ is the level coefficient of the learning rate. x[n] is calculated as the inverse of absolute difference between the chosen node and the incoming data, $x\left[n\right]=|d_{i}{-c|}^{-1}$ where $d_{i}$ is the input data and *c* is the value of the chosen node.
(4)$$y\left[n\right]=y\left[n-1\right]+k_{l}\times x\left[n\right].$$

As a result of the IIR filter, maturity cumulatively increases when the relevant node is triggered. If maturity exceeds a predetermined value, the spine node evolves into a cortex node. By controlling the $k_{l}$ coefficient, the speed of learning can be arranged. A higher $k_{l}$ value results in fast-evolving. As the cortical coding tree has a hierarchical tree structure, the probability of triggering a node at different levels of the tree is not equal. Therefore, the $k_{l}$ coefficient is set higher for high levels of the cortical coding tree.

The effective covering range parameter initially covers a wide range, which then narrows down to a predefined range of limited values as it is trained. There are two conditions for a node to be triggered or updated; first, the incoming data, i.e., the wavelet coefficient, should be within the values of the node’s covering range and also be the closest to the node’s value. If the spine node satisfies the conditions, it changes its value regarding Equation (Equation 6) and narrows down its covering range with the Equation (Equation 5) where *r* is the range parameter of the node, $r_{init}$ is the predefined initial range of nodes, $r_{limit}$ is the lower bound of a range, *w* is the pass count (weight) of the node and *l* is the level constant and *k* is the power coefficient of the weight parameter.
(5)$$r=\left\{\begin{array}{cc} \frac{r_{init}}{w^{k}l}, & r>r_{limit}\\ r_{limit}, & r\leq r_{limit}. \end{array}\right.$$

If conditions are not satisfied for the cortex and the spine nodes, a new spine node with the incoming wavelet coefficient value is generated. So, this means that if the incoming data are not well known or a previously similar value is not seen for the cortex node, then a spine node is generated for that unknown location. If this value is not an anomaly, and it will be seen more in the future, then this spine node evolves into a cortex node. As seen from Equation (Equation 5), well-trained nodes have narrow covering ranges, whereas less trained nodes have wider covering ranges. According to the two conditions, if a node is well trained (it means that it is triggered/updated many times), the covering range will be narrower and this increases the possibility of having more neighboring nodes where incoming data are dense. An example is presented in Figure 6. There are two nodes $c_{1}$ and $c_{2}$, at stage 4 in the figure, the input data are within the dynamic range of the node $c_{1}$ and therefore the $c_{1}$ node is updated, the value of the node decreases (the node moves downwards) and the range gets narrower and there becomes a gap between covering ranges of the $c_{1}$ and the $c_{2}$ nodes. Thus, a new node occurs at that gap in later stages. In order to maximize the entropy, we aim to obtain close to equally probable features. Generating more nodes dynamically at the dense locations of incoming wavelet coefficients results in more equally visited nodes in the cortical coding tree and increases entropy. The reason for limiting the range parameter to a predefined value is to prevent the range parameter not to be less than a certain value. Otherwise, a very well-trained node’s range parameter can get very close to 0, hence, this may result in generating neighboring nodes having almost similar values at the cortical coding tree. This causes a series of problems; firstly the algorithm becomes not memory efficient, extra memory is needed for new nodes having almost the same value. Secondly, the progeny of the neighboring values may not be as well trained as the original node, so at lower levels error rate increases. The cortical coding tree is a hierarchical tree and from the root to leaves, it follows the most similar path according to the incoming data and if the chain is broken at some level, the error rate may increase. Another reason for limiting the range parameter is to make the cortical coding tree convergent. Neighboring nodes range parameters can intersect, but this does not cause any harm to the overall development of the cortical coding tree because the system works in a winner takes all manner. The ranges can intersect but only the closest node to the incoming input value becomes the successive node. To summarize, $r_{limit}$ parameter can be considered the quantization sensitivity.

The training level of a spine node is controlled with a dissipation energy parameter. All the nodes in the cortical coding tree are dynamic; they adapt to the incoming data in their effective range and change their value (see Figure 6). The change in value is indirectly proportional to the train level of the node. The total change in the direction and the value of the relevant coefficients are determined by the incoming data and how well the node had been trained previously. The formula of the update of the node’s value ($\hat{c}$) according to the incoming data is presented in Equations (Equation 6) and (Equation 7), where *c* is the value of the node and, $x_{i}$ is the incoming ith wavelet coefficient, *w* is the pass count (weight) of the node, *l* is the level coefficient where l>1, *n* is the power of weight where 0.5≤n≤1 and *k* is the adaption control coefficient where 0<k<1
(6)$$\hat{c}=c+\frac{k\times c+\left(1-k\right)x_{i}-c}{{\left(w\times l+1\right)}^{n}}.$$

Grouping *c* parameter and coefficients, the formula can be simplified as in Equation (Equation 7).
(7)$$\hat{c}=c+\frac{\left(1-k\right)\left(x_{i}-c\right)}{{\left(w\times l+1\right)}^{n}}.$$

The weights represent the training level of the node, i.e., the pass count of wavelet coefficients via that node. Depending on the application type, the adaptation speed may vary. Smaller *k* values increase the significance of new coming input values rather than historical ones. For instance, applications like stream clustering mostly need faster adaptation. However, in normal clustering all values are important, so in our tests, *k* is chosen as 0.75. As cortical coding network has a hierarchical tree-like structure holding wavelet coefficients from low to high frequencies, the effect of change of node in each frequency level does not equally affect the overall signal. The change in low frequencies affects more than higher frequencies. To overcome and control this variance, level coefficient *l* is added to the equation. To control slower adaptation in low frequencies *l* can be chosen bigger comparing high frequency *l* data. In Figure 4, examples of different *k* values for dynamic change of the node value in Equation (Equation 7) is presented. Input data could be selected randomly, but on purpose to be more visible, a nonlinear function (sin(log(x)), x=1,…,49,50) is used to create the input data values. The initial value of the node is 0 and with each iteration, the change of the value of the node is presented depending on the *k* parameter in Equation (Equation 7). As a result of dynamic change of values in training, the cortex node values adapt to frequent incoming data values and this yields the increase of information entropy.

In order to better visualize the flow of the process of the cortical coding network training, an algorithm flowchart and pseudocode are presented Figure 5 and Algorithm 1. The flow chart shows how the cortical coding network is trained with the input wavelet coefficients for one frame of input data. At the start of each frame, the root node of the cortical coding tree is assigned as the selected node. The input vector is taken and from lowest to the highest frequency, the wavelet coefficients are trained in order. For the input coefficient in order, progeny nodes of the selected node are checked whether any node is suitable to proceed. There are two main conditions for a node to be suitable; the node’s value should be the closest to the incoming coefficient and input value should be within the covering range of the closest node. First, this control is checked among the cortex nodes of the selected node. If any suitable node is found then that node is updated and assigned as the selected node for the next iteration. If none of the cortex nodes are suitable, then the same procedure is carried out for the spine nodes of the selected node. If none of the spine nodes are suitable, then a new spine node with the input value is generated as a progeny spine node of the selected node. If a spine node is found suitable, then it is updated. If the maturity of the spine node exceeds the threshold, then it is evolved to a cortex node and this new cortex node is assigned as the selected node for the next cycle. If the spine node is not mature enough then the loop is ended as there will be no cortex node to be assigned as the selected node for the next iteration.
**Algorithm 1** Cortical Coding Network Training (for one frame)
    **Input**                 input wavelet coefficients vectors, $\left\{w_{1},w_{2},\ldots ,w_{d}\right\}$
    **procedure**
Cortical Coding Network (Cortex) Training          **for each** coefficient ($w_{i}$) in the input data frame **do**                **if** node has progeny cortex node **then**                      closest_node ← FindClosestNode(node, coefficient)                      **if** coefficient is in the range of the closest cortex node **then**                            node ← UpdateClosestNode(closest_node, coefficient)                            **continue**                      **end if**                **end if**                **if** node has spines **then**                      closest_spine ← FindClosestSpine(node, coefficient)                      **if** coefficient is in the range of the closest spine node **then**                            UpdateClosestSpine(closest_spine, coefficient)                            **if** closest_spine’s maturity > threshold **then**                                  node ← EvolveSpineToNode(closest_spine)                                  **continue**                            **else**                                  **break**                            **end if**                      **end if**                **end if**                GenerateSpine(node, coefficient)                **break**          **end for**
    **end procedure**


The complexity of the cortical coding algorithm is O(ndlogm), where *n* is the number of input vectors, *d* is the depth of the cortical coding tree (vector size), and *m* is the maximum number of progenies in the cortical coding tree. Since the cortical coding network is a sorted tree, finding the closest node in the algorithm is carried out with Binary Search. The complexity of Binary Search is given by O(logk), where *k* is the number of elements [29]. In practice, as the cortical coding tree is a hierarchical tree, the average progeny nodes in the cortical coding tree are much smaller than *m*, especially on lower level nodes which yields binary search operation to find the closest node on average becoming much faster on these nodes. So the algorithmic complexity can be considered as O(nd), discarding O(logm). Additionally, there are many break commands in the algorithm which end the loop earlier. For these reasons, even though the time complexity seems bigger than Birch, the algorithm runs much faster than it.

### 2.4. Entropy Maximization

The major aim of the cortical coding algorithm is to maximize the information entropy at the level of the leaves of the cortical coding tree, i.e., to generate a codebook that has equally probable clustered features along each path from the root to the leaves. The process of entropy maximization is carried out at each consecutive level of the tree separately until they converge to a maximum vector feature (see Appendix C). The two types of nodes present in the cortical coding tree are cortex and spine nodes (Figure 3). A spine node is the progeny of the cortex node and is the candidate to be the next cortex node. This means that a well-trained spine evolves into a cortex node. Unlike a cortex node, a spine does not have any progeny node. Each spine has a maturity level that dictates its evolution to be a cortex node. When a relevant spectral feature passes through a cortex node and if there is no progeny cortex node yet within the dynamic range of the previously created, then the spines of the node are assessed. If the wavelet coefficient is not within any of the dynamic ranges of the spine nodes, then a new spine node with the incoming wavelet coefficient is generated. If the relevant spectral feature is within the effective dynamic range of previously created spines, then the spine node in the range is triggered and updated. If the maturity of a spine node passes through a predetermined threshold, then the spine node evolves to a progeny cortex node. A numerical example is presented in Figure 6, steps 1 through 6. Here, the formation of the nodes and the evolution of the spines into tree nodes take place according to the incoming data. The cortical coding network is generated through: (1) A *l*th level node in the cortical coding tree has neither spine nor cortex nodes when first formed. (2) After a certain period of time, the node forms two spine nodes ($s_{1}$, $s_{2}$) and two progeny nodes with data $c_{1}$ and $c_{2}$ with covering ranges of $r_{1}$ and $r_{2}$, respectively. (3) A new spine, $s_{3}$, is created when a signal arrives outside the range of any cortex or spine node. (4) The arrival of the new signal *w* triggers an update of the range and the data values. While the node slightly changes its position, the new center becomes ${\overset{´}{c}}_{1}$ and the covering range narrows down to ${\overset{´}{r}}_{1}$, which continues as the training level of the node increases eventually causing the formation of a gap between the coverage ranges of $c_{1}$ and $c_{2}$. (5) When the new signal *w* arrives outside the covering ranges of nodes but within the covering range of $s_{2}$ spine node, then this node is updated, i.e., the center of $s_{2}$ adapts to the signal with its range narrowed down. As the increased maturity level of $s_{2}$ exceeds a threshold value, a new spine node forms as a progeny cortex node. (6) A signal *w* similar to the previous one arrives, again $s_{2}$ node is updated, as the maturity level of $s_{2}$ node surpasses the threshold, the $s_{2}$ node eventually evolves to a cortex node with its center at $c_{3}$, covering a new range $r_{3}$. The mathematical description of evolving of spines into cortex nodes is explained earlier in Section 2.3).

Entropy is maximized when an equal probability condition is reached. To have an efficient vector quantization the aim is to generate equally probable visited leaves of the tree. Shannon’s information entropy *H* is calculated as follows [15]:(8)$$H\left(X\right)=-\sum_{i=1}^{n}P\left(x_{i}\right)log_{b}P\left(x_{i}\right),$$
where *X* is a discrete random variable having possible outcomes $x_{1}$ to $x_{n}$ and the possibility of these outcomes are $P(x_{1})$ to $P(x_{n})$, *b* is the base of the logarithm and it differs depending on the applications. Cortical coding network is dynamically developed and aims to maximize entropy at the leaves of the network. At each level of the tree, the rules are applied. More frequent and probable data generate more nodes comparing the severe areas. So at each level, it aims to maximize entropy. However, the main goal is to have equal probable features at the leaves of the cortical coding tree. Consecutive network levels yield an increase in entropy (see Appendix C). An example of entropy maximization in one level is given in the Figure 7. Three random normally (Gaussian) distributed signals with means 0, −10, 10 and standard deviation 5, 3 and 2 are concatenated to generate the input signal. Each signal has $10^{5}$ samples so the generated signal has $3\times 10^{5}$ samples. The histogram of the signal is given in Figure 7b. The entropy maximization of the cortical coding algorithm compared with K-means, uniform distributed nodes and the theoretical maximum entropy line are presented in Figure 7a. Uniform distributed nodes and the theoretical maximum entropy are added to comparison in order to have a baseline and upper limit comparison. The theoretical max entropy is calculated with Shannon’s entropy (*H*) where the logarithmic base (*b*) is equal to the number of nodes, so it is 1.0. Uniform distributed nodes given in Figure 7a are generated by calculating the range of the distribution (almost −20 to 20, see Figure 7b *x* axes) and inserting equally distanced nodes in the calculated range. As expected, the visiting probability of the uniform distributed nodes mimics the shape of the histogram of the signal with H=0.8627. The cortical coding network (Cortex) entropy (H=0.9764) is slightly better than K-means entropy (H=0.9632). The example is given in one level of the tree. In consecutive levels of the tree, entropy is increased by each level. The node generation of the cortical coding algorithm is convergent, depending on the range limit ($r_{limit}$) parameter. This parameter as mentioned earlier defines the minimum value of the effective range parameter to limit dynamic node generation. An example is presented in Figure 7c, as the training epoch increases the total number of nodes increase in a convergent manner. Range Limit parameter directly affects the number of nodes generated. When it is small, it means that a well-trained node has a narrow effective range and the close neighborhood of this node is not in the node’s effective range. So, more nodes are generated in the neighborhood. In Figure 7d the effect of range limit parameter is presented. The cortical coding network is trained with the same input data but with varying $r_{limit}$ parameters. The change in visiting probability according to $r_{limit}$ parameters can be seen from the figure. The information entropy results (*H*) in Figure 7d are presented in $log_{2}$ scale.

### 2.5. Comparison of Algorithms for Codebook Generation

There are various types of clustering categorization in the literature. Xu, Dongkuan, and Tian made a comprehensive study on clustering types and in their research, they divide clustering methods into 19 types [30]. Rai, Pradeep, and Sing divide clustering algorithms into four main groups, partition-based clustering, hierarchical clustering, density-based clustering and grid-based clustering [31]. A common way of categorizing clustering algorithms is presented in Figure 8.

In order to assess the utility of the cortical coding based algorithm in a broader context, its performance was compared with those of the commonly used clustering algorithms in vector quantization. As the representative clustering models, we choose GMM/EM, K-means, and Birch/PNN algorithms covering most categories, as shown in Figure 8. Additionally, these models were chosen such that they would each generate a centroid point and that a number of cluster size (quantization number) could be set as an initial parameter. The algorithms were not selected from density-based and grid-based models as they would not satisfy these criteria. From partition-based clustering, K-means is chosen as it is a vector quantization method that partitions a certain number of points (n) into (k) clusters. The algorithm tries to match each point to the nearest cluster centroid (or mean) iteratively. In each iteration, the cluster centroid changes whether new points are assigned or not. The algorithm converges to a state where there is no cluster update for points. The main objective in this model is to end up in a state where in-class variances are maximum, and between-class variances are minimum. K-means is one of the most popular partition-based clustering algorithms and it is easy to implement, fairly fast, and is suitable for working with large datasets. There are many variations of K-means algorithm to increase the performance. The improvements can be classified as better initialization, repeating K-means various times, and the capability of combining the algorithm with another clustering method [32,33,34]. The standard model which is also called Lloyd’s algorithm aims to match each data point to the nearest cluster centroid. The iterations continue until the algorithm converges to a state where there is no cluster update for points [35]. The algorithm converges to the local optimum, and the performance is strictly related to determining initial centroids. K-means is considered to be an NP-hard problem [36]. The time complexity of the standard K-means algorithm is O(nkdi) where *n* is the number of vectors, *k* is the number of clusters, and *d* is the dimension of vectors while *i* is the number of iteration. K-means is commonly used in many clustering and classification applications such as document clustering, identifying crime-prone areas, customer segmentation, insurance fraud detection, bioinformatics, psychology, and social science, anomaly detection, etc. [34,37,38]. The pseudocode of the K-means algorithm used is presented in Algorithm 2.
**Algorithm 2** k-means
    **Input**          X     input vectors, $\left\{{\vec{x}}_{1},{\vec{x}}_{2},\ldots ,{\vec{x}}_{n}\right\}$          k     number of clusters
    **Output**          C     centroids, $\left\{{\vec{c}}_{1},{\vec{c}}_{2},\ldots ,{\vec{c}}_{k}\right\}$
    **procedure**
K-means          **for** i←1…k **do**                $c_{i}\leftarrow$ Random vector from *X*          **end for**          **while** not converged **do**                Assign each vector in *X* to the closest centroid *c*                **for each** c∈C **do**                      c← mean of vectors assigned to *c*                **end for**          **end while**
    **end procedure**


Birch was chosen as the second model for comparison from among the hierarchical clustering algorithms as it is one of the most popular methods for clustering large datasets and has very low time complexity yielding in fast processing. The model’s time complexity is O(n) and a single read of the input data is enough for the algorithm to perform well. Birch consists of four phases. Firstly, it generates a Clustering Feature tree (CF) in phase 1 (see Algorithm 3). A CF tree node consists of three parts; a number, the linear sum, and the square sum of data points. Secondly, phase 2 is optional and it reduces the CF tree size by condensing it. In Phase 3, all leaf entries of the CF tree are clustered by an existing clustering algorithm. Phase 4 is also optional where refinement is carried out to overcome minor inaccuracies [39,40]. In Phase 3 of the Birch algorithm, an agglomerative hierarchical method (PNN) is used as the global clustering algorithm. It is one of the oldest and most basic vector quantization methods providing highly reliable results for clustering. PNN method starts with all input data points and merges the closest data points and then deletes these until a desired number of data points is left [7,41]. The drawback of the algorithm is its complexity, i.e., the time complexity of agglomerative hierarchical clustering is high and often not even suitable for medium size datasets. The standard algorithm has $O(n^{3})$ but with refinements, the algorithm can work $O(n^{2}logn)$. The pseudocode of PNN is presented in Algorithm 4.
**Algorithm 3** Birch
    **Input**          X     input vectors, $\left\{{\vec{x}}_{1},{\vec{x}}_{2},\ldots ,{\vec{x}}_{n}\right\}$          T     Threshold value for CF Tree
    **Output**          C     set of clusters
    **procedure**
BIRCH          **for each** ${\vec{x}}_{i}\in X$ **do**                find the leaf node for insertion                **if** leaf node is within the threshold condition **then**                      add ${\vec{x}}_{i}$ to cluster                      update CF triples                **else**                      **if** there is space for insertion **then**                            insert ${\vec{x}}_{i}$ as a single cluster                            update CF triples                      **else**                            split leaf node                            redistribute CF features                      **end if**                **end if**          **end for**
    **end procedure**


**Algorithm 4** Pairwise Nearest Neighbor.
    **Input**          X     input vectors, $\left\{{\vec{x}}_{1},{\vec{x}}_{2},\ldots ,{\vec{x}}_{n}\right\}$          k     number of clusters
    **Output**          C     centroids, $\left\{{\vec{c}}_{1},{\vec{c}}_{2},\ldots ,{\vec{c}}_{k}\right\}$
    **procedure**
PNN          **while** N>k **do**                $({\vec{c}}_{a},{\vec{c}}_{b})\leftarrow$ FindNearestClusterPair()                MergeClusters$\left({\vec{c}}_{a},{\vec{c}}_{b}\right)$                N←N−1.          **end while**
    **end procedure**


From model-based clustering, Gaussian Mixture Model (GMM) with Expectation Maximization (EM) was chosen for comparison in the present work. GMM is a probabilistic model in which the vectors of the input dataset can be represented as a mixture of Gaussian distributions with unknown parameters. Clustering algorithms, such as K-means, assume that each cluster has a circular shape, which is one of their weaknesses. GMM tries to maximize the likelihood with the EM algorithm and each cluster has a spherical shape. While K-means aims to find a *k* value that minimizes ${\left(x-\mu \right)}^{2}$, GMM aims to minimize ${\left(x-\mu \right)}^{2}/\delta^{2}$ and has a complexity of $O(n^{2}kd)$. It works well with real-time datasets and data clusters that do not have circular but elliptical shapes, thereby increasing the accuracy in such data types. GMM also allows mixed membership of data points and can be an alternative to fuzzy c-means, facilitating its common use in various generative unsupervised learning or clustering in a variety of implementations, such as signal processing, anomaly detection, tracking objects in a video frame, biometric systems, and speech recognition [12,42,43,44]. The pseudocode is presented in Algorithm 5.
**Algorithm 5** GMM
    **Input**          X     input vectors, $\left\{{\vec{x}}_{1},{\vec{x}}_{2},\ldots ,{\vec{x}}_{n}\right\}$          k     number of clusters
    **Output**          C     centroids, $\left\{{\vec{c}}_{1},{\vec{c}}_{2},\ldots ,{\vec{c}}_{k}\right\}$, $c_{i}=\left(\phi_{i},\mu_{i},\sigma_{i}^{2}\right)$
    **procedure**
GMM          **for** k←1…K **do**                $\phi_{k}\leftarrow \frac{1}{K}$                $\mu_{k}\leftarrow$ Random vector from *X*                $\sigma_{k}^{2}\leftarrow$ Sample covariance          **end for**          **while** not converged **do**                **for** k←1…K,i←1…N **do**                      $\gamma_{ik}=\frac{\phi_{k}N\left(x_{i}|\mu_{k}\;\sigma_{k}^{2}\right)}{\Sigma_{j=1}^{K}\phi_{j}N\left(x_{i}|\mu_{k},\;\sigma_{k}^{2}\right)}$                **end for**                **for** k←1…K **do**                      $\phi_{k}=\Sigma_{i=1}^{N}\frac{\gamma_{ik}}{N}$                      $\mu_{k}=\frac{\Sigma_{i=1}^{N}\gamma_{ik}x_{i}}{\Sigma_{i=1}^{N}\gamma_{ik}}$                      $\sigma_{k}^{2}=\frac{\Sigma_{i=1}^{N}\gamma_{ik}{\left(x_{i}-\mu_{k}\right)}^{2}}{\Sigma_{i=1}^{N}\gamma_{ik}}$.                **end for**          **end while**
    **end procedure**


The performance of the cortical coding algorithm was compared in terms of execution time and distortion rate with the above listed commonly used clustering algorithms, each of which can create a centroid point suitable for vector quantization. As discussed, while K-means is a popular partition-based fast clustering algorithm, Birch is a hierarchical clustering algorithm that, even though it is slower than cortical coding method in terms of execution time, has an advantageous coding performance with low time complexity. GMM, as well as Birch, are model-based clustering algorithms and both are effective in overcoming the weakness of K-means. Each of the algorithms used for comparison, therefore, has different methods to solve the vector quantization and clustering problem, each having pros and cons. More specifically, K-means performance is related to initial codebook selection but has local minimum problem. In contrast, hierarchical clustering does not incorporate a random process; it is a robust method, but the algorithm is more complex than the K-means model. Agglomerative hierarchical clustering algorithms, like PNN, usually give better results in terms of distortion rate than K-means and GMM but their drawback is in time complexity. GMM produces fair distortion rates, better than K-means as it solves the weakness of this method with spherical clusters. Although GMM is a fairly fast algorithm when vector size is small, its drawback is time complexity, which increases when high dimensional vectors are present in the dataset. The common denominator of these algorithms, however, is that each uses a look-ahead buffer, which means they require the knowledge of all input vectors in the memory at the start of the process. This drawback in conventional algorithms that substantially slows down execution process, is overcome in the new model; the cortical coding algorithm works online, the key difference with K-means, Birch/PNN and GMM. This fast execution is enabled by a brain-inspired process, in which the input vectors are recorded to the cortical coding network one by one in a specific order and which then eventually become a sorted n-array tree (Figure 1, Figure 3 and Figure 6).

To make the base comparisons among the algorithms, C++ is used for developing algorithms. C++ is a low-level programming language and is commonly used in many applications working on hardware, game development, OS, and real-time applications, where high performance is needed, etc., and it is especially chosen for performance reasons. We aim to compare the pure codes of the algorithms used in comparison. Commonly used libraries, however, may make use of extra optimizations in codes, some parallel processes, or extended instruction sets, which may yield performance improvements. Therefore, aiming to develop custom codes for all comparison algorithms, we wrote K-means, Birch, PNN, and GMM in C++. Custom GMM code has some problems with high dimension matrix inverse and determinant calculations; therefore, for some dataset, the convergence of the algorithm took extremely long and the quantization performance was only fair. As a result, we decide to use a commonly used machine learning C++ library for GMM. Mlpack is a well-known and widely used C++ machine learning library [45]. We, therefore, used GMM codes in mlpack library, but, discovered that the execution time was very high. We believe that this is because the library code is not optimized for large number of clusters. We also tested PyClustering library, in which there are various types of clustering algorithms [46]. Although most of these algorithms are written both in C++ and Python, unfortunately GMM is not one of these algorithms. Since Python is an interpreted language and usually uses libraries that were implemented in C++, it would always introduce an additional cost in execution time that is spent for calling the C++ libraries within the Python. It also has callback overhead even for libraries that were implemented in C++. Even though Python codes are slower than C++ codes, the scikit-learn is a well-optimized library and, thus, GMM results are much better (in terms of less execution time and less distortion rate) compared to mlpack library and our custom developed algorithm [47]. As a result, we used scikit GMM codes in the comparison process discussed in the main text. In spite of these advantages of C++, to make further comparisons of the execution time and distortion for a even broader implementations, we also develop all algorithms in Python. Being an interpretable language, since it is not compiled, Python is slower than C++. Therefore execution time increases in Python versions but distortion rates stay the same. The data generated in this work, i.e., simple waves and Lorenz chaotic data, have been written in Python.

## 3. Results

The comparison of training and test scores is crucial to determine the generalization performance of the algorithms. In machine learning, the generalization term is an essential concept as it shows the ability of the model to adapt the previously unseen data with the same distribution with trained data. More specifically, in an ideal world, the performance of a given model with training and testing datasets preferably be similar [48]. Using two different datasets, the performances of the algorithms were tested in two different metrics, i.e., distortion and execution time. The significance of distortion is that it represents the success of vector quantization as it is the measure of quantization error. Small distortion reflects that quantized signals are highly similar to those of the original. The importance of the execution time is that as it decreases, then the overall algorithm performs with a related energy gain. Each result is presented with trained and untrained data to compare the differences in the performance among the four different algorithms. Test dataset is generated with the same distribution of the training data. Even though the test dataset has the same distribution, it is different than the train dataset and not previously seen by the algorithms at the training stage. Therefore, the test evaluates whether the algorithms memorize and overfit the train dataset or a more generalized solution may be found. Here, the distortion is measured by the root-mean-square error (RMSE) of the original and quantized data while the execution time of all algorithms for training datasets is presented. All of the numerical experiments were carried out for all the algorithms by using the same computer for a fair assessment of the execution times. CPU clock time of the computer is used to measure the execution time which allows the quantification of the execution times with microsecond sensitivity. For each algorithm, there are six tests with different sizes; e.g., for 16K→330, initially there are 16,000 vectors that are quantized to 330. The training results show that the distortion is better for Birch and GMM rather than cortex (cortical coding method) and K-means models. It can also be noticed that the cortex and K-means results are almost the same. However, when the input vector size is small, the K-means performs slightly better and vice versa, i.e., cortical coding model performs better when the input vector size increases (see Figure 9c). Once the codebooks are generated with the trained datasets, the new signals with the same characteristics are regenerated for untrained datasets. In this case, the cortical coding model performs better than K-means for all basic wave tests while Birch and GMM still perform better than the cortex, although the differences in the results of the cortex and Birch and GMM become smaller. The conclusion from these results is that the cortical coding model has better generalization performance, since it adapts untrained datasets better than the traditional algorithms.

In order to compare the performance of the four algorithms, both the execution times and the distortions need to be evaluated simultaneously. Because of the very large differences in execution times measured in the unit of seconds, they are presented in $log_{10}\left(time\right)$ scale (Figure 9a,b). The most dramatic result is that the execution time of the cortex far outperforms all other algorithms used, a second versus minutes, or hours, respectively.

To establish further performance tests of the cortical coding model compared to the chosen traditional algorithms, we also used a different type of dataset, specifically, x series of Lorenz chaotic data using both trained and untrained datasets. Here, the first test was performed to evaluate the distortion rate, for which six different tests with various initial vector sizes are presented. In terms of the distortion rate, Birch performs almost the same with GMM, which performs better than K-means and cortical coding models. K-means results are slightly better than the cortical coding model for the trained dataset. For the untrained dataset, however, the distortion in the cortical coding model remains almost the same as the trained data results while the distortion rates of all the tested models become worse (see Figure 9d). Having similar distortion rates with trained and untrained data is a significant advantage of the cortical coding model. The results show that the cortical coding algorithm generalization is comparably better than all traditional algorithms. In Figure 9b, the execution time performances of the algorithms are presented when the Lorenz dataset is used, displaying that the results are similar to the first dataset when basic waves are used, i.e., the cortical coding model suppresses the other algorithms in execution time in codebook generation. More specifically, the cortical coding algorithm in using both datasets, approximately, takes about a second while Birch model takes several hundreds of seconds, K-means ends up in minutes, and GMM in hours, using the same datasets (see Section 2.5 in Methods for more detailed description of codebook generation using these algorithms).

## 4. Discussions

In most machine learning applications, training is the most time-consuming step [49]. Cortical coding algorithm has a significant advantage in reducing the training time (see Figure 10a). There could be substantial gains in each of the commonly used algorithms in cases where the vector quantization is carried out using the cortical coding methodology. A metric for the gain may be determined by combining the execution time and the distortion. A comparison of the cortex vs traditional algorithms in terms of a gain metric, measured in execution time and distortion rate, according to the equation in the inset (Figure 10b). Here, the gain (in log scale) represented by colored arrows indicates the min, max, and average (asterisk) values versus the traditional algorithms (all based on the results of basic wave tests). On average, the gain in cortical coding algorithm is better than Birch (≈×170), K-means (≈×700), and GMM (≈×4700), i.e., there is a substantial gain of cortical coding algorithm versus the algorithms used for comparison (colored vertical arrows). When comparing other important performance parameters, the cortical coding algorithm also fares significantly better than the common frequently used algorithms. This is shown in the radar graph in Figure 10c, and also presented numerically in Appendix D. For all tests, the values at the periphery represent better performance in the radar graph; therefore, all results are normalized according to the best performance among the four algorithms. Only train and test fidelity have been taken into consideration together to have an improved comparison and normalized as the best performance of the training results (i.e., GMM). As discussed, generalization is the key phenomenon in machine learning applications as it defines the ability of the developed model to react to previously unknown data. For comparisons, generalization is taken as the ratio of test-to-train, and those results closer to the value of 1 are better. As shown in the graph, the generalization performance of the cortical coding algorithm surpasses all others. It should be emphasized that if the trained performance of the model is decent but the algorithm does not perform well for a new dataset, then its generalization error is high, showing simply how well an algorithm performs for the new data with the same distribution as the trained set. In the graph, the execution time is presented in the $log_{10}$ scale, where, again, the cortical coding algorithm outperforms all other methods. Although the train and test results do not seem favorable, the generalization performance of the cortical coding algorithm surpasses others, implying that it is a more generalized solution. Execution time ratio (log scale) is normalized according to the best result, in which the cortical coding algorithm surpasses the other methods. The higher value means better scalability where the cortex performs as well as the best performing algorithms. Finally, a significant feature of the cortical coding algorithm is its scalability performance, i.e., the ability of an algorithm to maintain its efficiency when the workload grows; here the cortical coding algorithm performs as well as the best performers (scalability results are taken from [30]).

The feature extraction performance of the proposed cortical coding algorithm in codebook generation is almost the same as the compared methods, i.e., hierarchical clustering, K-means, and GMM, based on the same dataset. While the hierarchical clustering performance is the best out of four algorithms in the trained dataset, the cortical coding approach converges faster in the case of an untrained dataset, a strong indication of its unique potential of widespread utility. We emphasize, again, that there are two key advantages of the cortical coding algorithm. First, in terms of time complexity, the execution time is much faster. Even though the operations are based O(ndlogm) (in the worst-case scenario), in practice, the algorithm works even much faster. This is due to the fact that, while the operations are executed vectorally in the execution of traditional algorithms, this is accomplished in a scalar manner in the construction of the cortex in a hierarchical network architecture. Therefore, the cortical coding approach allows one to have many break loop commands in the algorithm so that in practice the execution time is significantly faster. Secondly, the algorithm works in an online manner and does not use a look-ahead buffer. In fact, this is the key advantage, for example, for a continuous learning system or stream clustering. These two significant advantages of the algorithm yield significant opportunities in practice and could facilitate unique application opportunities as it can be used, e.g., as a feature extractor and signal encoder, or as a new methodology in pattern recognition.

The practical significance of the cortical coding algorithm is that it may be implemented in a multi-layer data encoder/decoder system to substantially enhance the performance in potent future implementations. Multi-level encoder and decoder cortical coding architecture is schematically shown in Figure 11. In this representation, there are three encoder and three decoder layers. During the network formation, the encoder and decoder parts in the architecture are mirrored with respect to each other and the communication is bidirectional so that each side of the channel can potentially work as both the encoder and the decoder. The first encoder layer is the feature extractor in vector quantization, described in this work. Here, again, the features of the input data are quantized with minimum dissipation and maximum entropy principle.The first encoder, therefore, aims to maximize the entropy of the fixed size input vectors. The leaves of the first layer encoder network are the input for the second encoder layer, where it codes the consecutive first encode layer indices.The second encode layer, therefore, is the collector; it encodes most frequently triggered consequent leaves of the first encoder layer. The leaves of the second encoder tree have a wider resolution. Similarly, frequently triggered consecutive leaves of the second encode layer are encoded at the third encode layer, where the outputs are the symbols. For example, if the output of the first encode layer network is ‘letters’, the second encode layer codes ‘syllables’ and the third encode layer then codes ‘words’. So, from an unknown input space, regarding the incoming features of the signals, the frequently occurring features are separated and the most probable incoming features are coded together to increase the performance. Second and third layers code in the symbol space. Here, the symbols are sent via the communication channel to receiver side. The decoder layers in the receiver side are exactly the same in reverse order. Similar to the way the reverse order symbols are partitioned into smaller symbols, the smaller symbols are decoded to feature indices which, in turn, are decoded to generate wavelet coefficients. The inverse transformed wavelet signals are, then, denormalized and the decoded signal is, thereby, created. Depending on the type and characteristics of a given application, coding performance or compression ratio expectation, the coded signal that can be transferred from the channel can readily be varied. For instance, in applications where the encoding error is critical and should not exceed a predefined threshold value, the encoding and decoding can be carried out with the combination of earlier layers. If all encode and decode levels have depth of four, then one 1st Decode layer index consists of four 2nd Decode layer indices and one 2nd Decode layer index consists of four 3rd Decode layer indices. Therefore, instead of one 1st Decode layer index, for instance, three 2nd Decode layer indices and four 3rd Decode layer indices can be used to decrease the error rate, since lower layers have less error rate and compression ratio. The overarching goal here would be to minimize the error rate and maximize the compression ratio. Consequently, encoding and decoding comprising a combination of all layers will decrease the compression ratio towards achieving a decoded signal with less error.

## 5. Conclusions

Inspired by energy–entropy relations in biological neural networks, we developed a cortical coding algorithm that generates a fast codebook. As we show here, the hierarchically structured cortical coding network is created in real-time from the features of the applied data without the initial structural definition of the interconnection-related hyperparameters and, therefore, represents a novel approach different from conventional artificial neural networks. The computational efficiency and the generalization performance of the generated codebook-equivalent cortical network are superior to the commonly used clustering algorithms, including K-means clustering, Birch, and GMM methods. Since the computational complexity of the cortical coding method has linear characteristics like Birch, it enables the generation of codebooks with large datasets. As demonstrated, the execution time is reduced drastically, in seconds versus minutes or even hours in some cases for clustering problems. More significantly, as the input data size increases the computational time efficiency increases with respect to the other common methods. The cortical coding approach, therefore, offers a new basis for ML platforms potentially paving the way towards real-time practical, multifaceted, and complex operations including reasoning and decision making. Although only vector quantization property was demonstrated in the presented work, the cortical coding algorithm has a real potential in a wide variety of applications, e.g., temporal clustering, data compression, audio and video encoding, anomaly detection, and recognition, to list a few. Looking forward, the codebook generation model presented herein is the first step towards a more generalized scheme of connectome-based machine intelligence model that covers all layers of a cognitive system from sensory functions to the motor activities, which the authors are currently pursuing and will report in the near future.

## Figures and Tables

**Figure 1 entropy-24-01678-f001:**
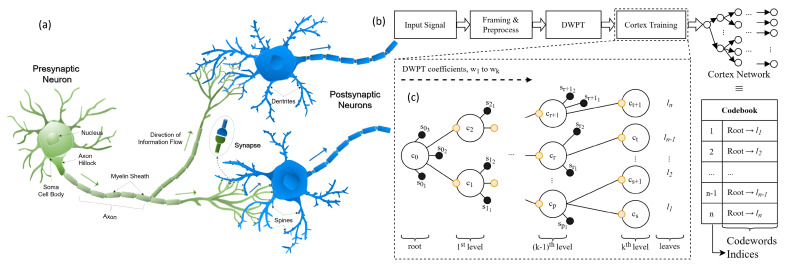
Biological neuron and general overview of the cortical coding algorithm. (**a**) A general representation of the neural network which forms a cortex in a biological brain (e.g., rats). (**b**) A block diagram representing the generalized flow of the brain-inspired cortical coding network formation. (**c**) Hierarchical structure formation of the brain-inspired cortical coding network. See text for details.

**Figure 2 entropy-24-01678-f002:**
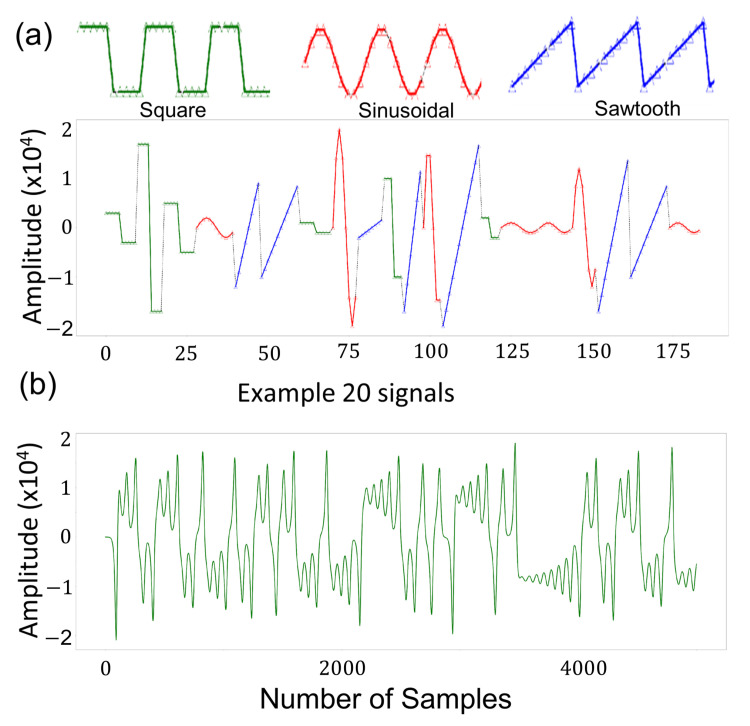
The datasets for testing the algorithms. Two different datasets were generated constituting the input data. (**a**) Basic waves dataset that includes three basic signals, sinus, square, and sawtooth. (**b**) Lorenz chaotic dataset, see text for details.

**Figure 3 entropy-24-01678-f003:**
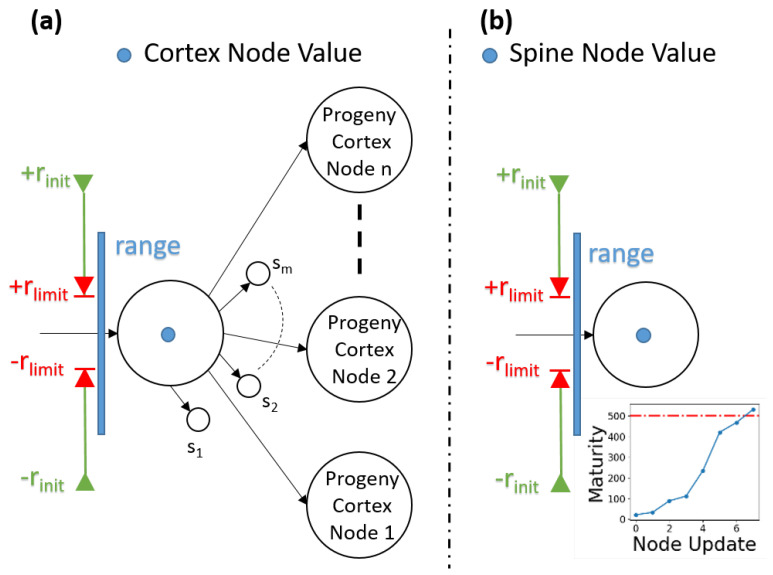
Nodes of the Cortical Coding Tree. (**a**) Cortex node may have progeny cortex and nodes, and it operates under three main variables; range parameter, node value, and pass count. Cortex node’s parent is also a cortex node. (**b**) Spine nodes are candidates that would turn into cortex nodes. Spine nodes do not have any progeny nodes and their parent is a cortex node. Unlike a cortex node, therefore, a spine node has a maturity parameter, which controls the evolution of spine into a cortex (see inset). This process of evolution nodes mimics that if node formation in a biological brain (see Figure 1a).

**Figure 4 entropy-24-01678-f004:**
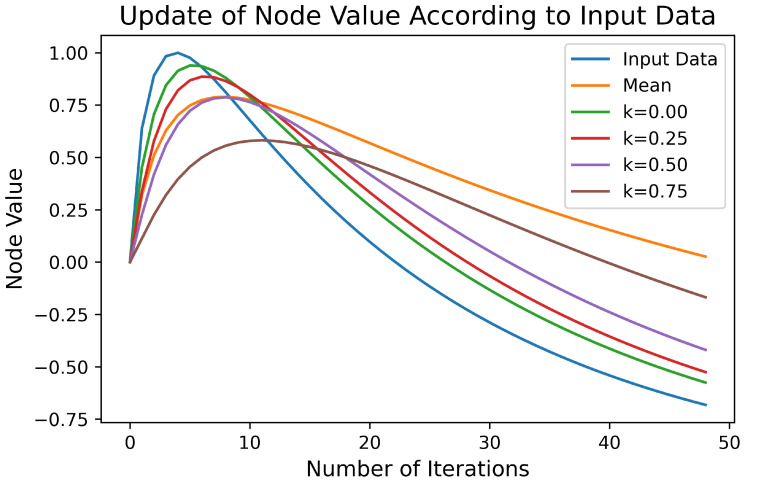
An example of updating of the node values according to different *k* values in Equation (Equation 7). Here, the blue line shows the values of the incoming data. The mean of input values is shown with orange line. The updating of a node’s value is given for four different *k* values, according to Equation (Equation 7). Node values adapt faster to incoming data for smaller *k* values. The power of the pass count parameter (*n*) is chosen as 0.5 for all cases. Smaller *n* values in Equation (Equation 7) provide faster adaptation, similar trend as in *k* values. Depending on the application types, both *n* and *k* values can be optimized. For clustering, for example, slow adaptation parameters are chosen; therefore, the significance of the historical values become greater than those of the newly coming values.

**Figure 5 entropy-24-01678-f005:**
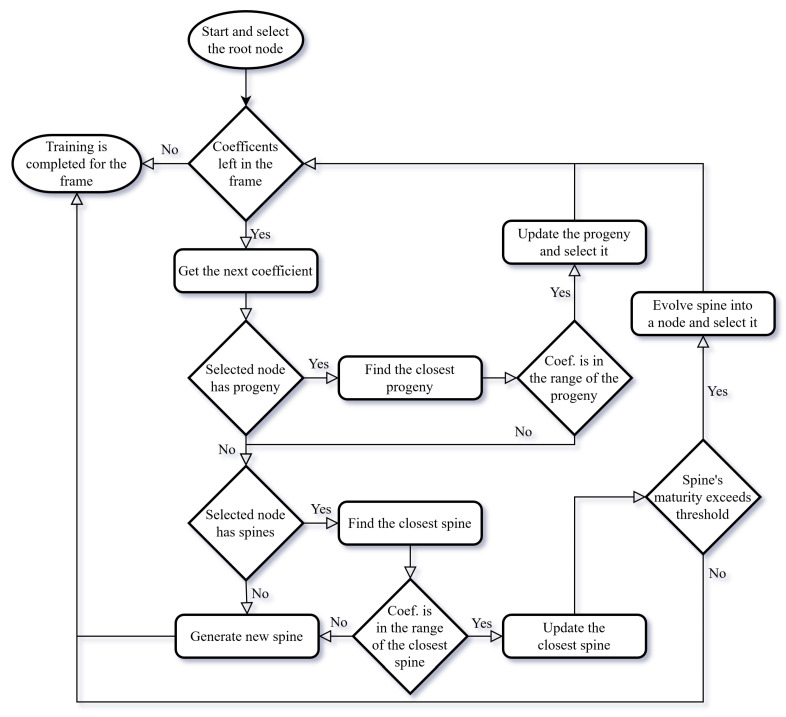
The flowchart of the cortical coding network training algorithm.

**Figure 6 entropy-24-01678-f006:**
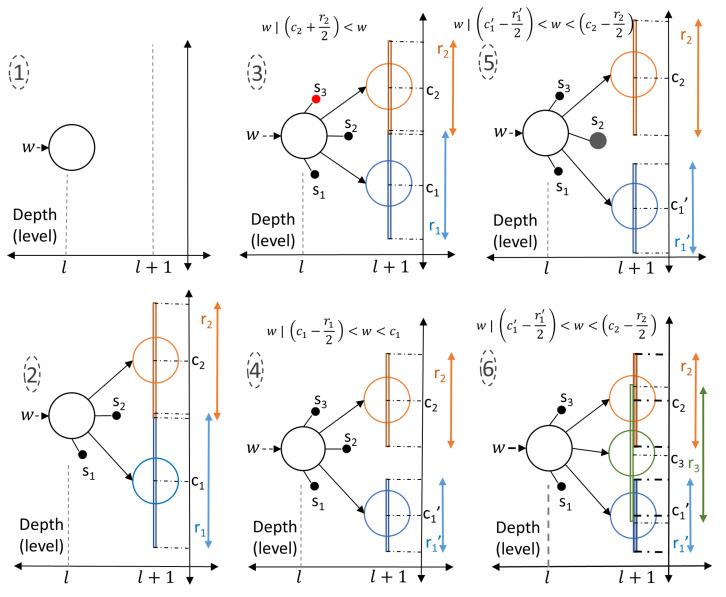
Dynamic generation of cortical coding network through entropy maximization. See text for details.

**Figure 7 entropy-24-01678-f007:**
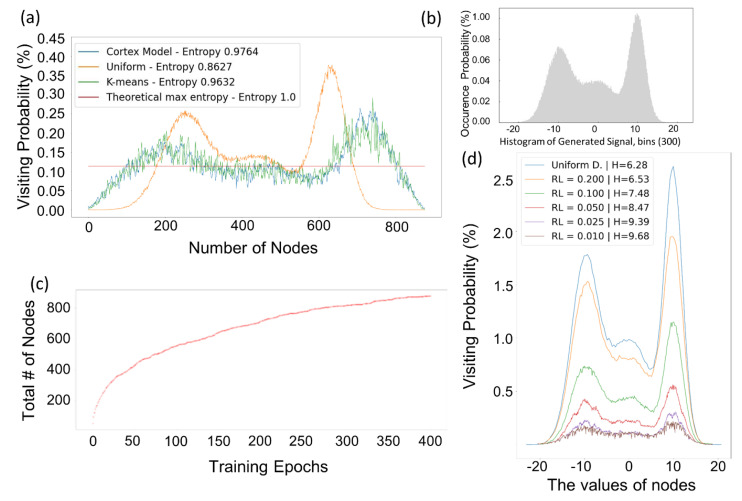
Entropy maximization and convergence. (**a**) In this example, three Gaussian distributed signals are randomly concatenated to generate the input signal. The signals have mean values of 0, −10, 10, with the standard deviation of 5, 3, and 2, respectively. The input dataset has real values varying within the range of −20 to 20. Each signal has 100,000 sample data points, therefore, the concatenated input signal has 300,000 sample data points. (**b**) Histogram of the generated three Gaussian signals given in (**a**). (**c**) The number of nodes vs training epochs in a cortical coding tree. The cortical coding algorithm converges as the number of epochs is increased. (**d**) The change of visiting probability of nodes vs values of nodes depending on the range limit (RL) parameter. The value of RL is a predefined parameter used for setting the minimum value to determine how close two cortex nodes can be. Here, cortical coding algorithm was run with five different RL values. The visiting probability of uniform distributed nodes has been included for baseline comparison. The entropy values (H) are given in $log_{2}$ base.

**Figure 8 entropy-24-01678-f008:**
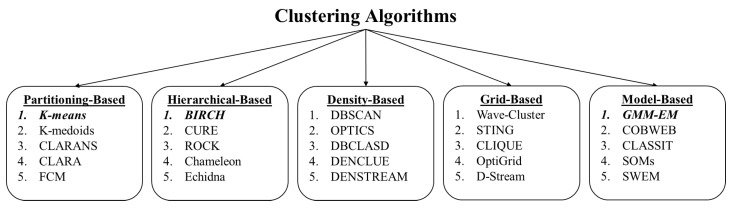
A common taxonomy of clustering algorithms. Popular algorithms from each cluster type that are suitable for comparison in this work are highlighted.

**Figure 9 entropy-24-01678-f009:**
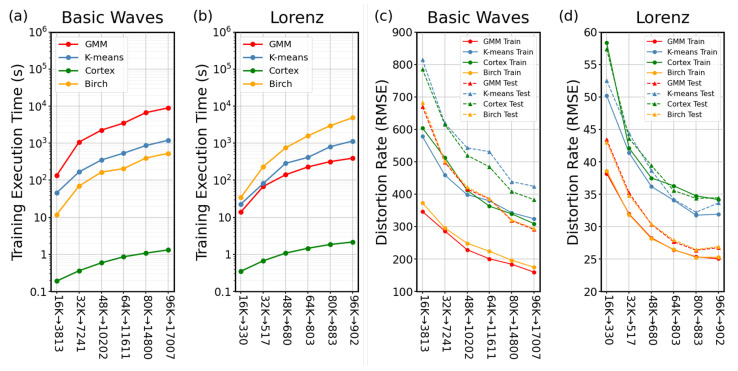
Test results of execution time and distortion rates. For each algorithm, there are six tests with different sizes; e.g., for 16K→330, initially there are 16,000 vectors that are quantized to 330. (**a**,**b**) Training Execution Time using both datasets, respectively. The performance of the cortex surpasses those of the other algorithms. (**c**,**d**) Distortion rate performance of the algorithms in training and testing, using (**c**) Simple wave and (**d**) Lorenz chaotic datasets.

**Figure 10 entropy-24-01678-f010:**
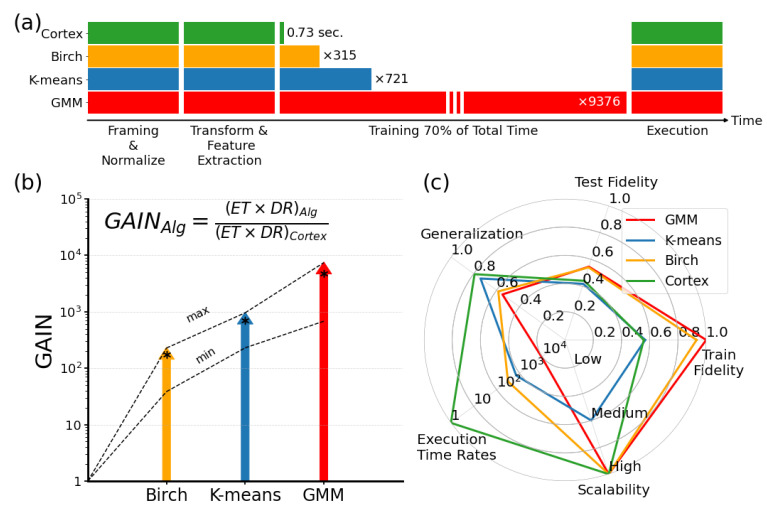
The practical significance of the cortical coding algorithm when considering execution time and distortion rate together. (**a**) The training is the most time-consuming step in all general machine learning algorithms, where the cortical coding algorithm has the best performance, i.e., fraction of second vs. minutes and hours. (**b**) A comparison of the cortex vs. traditional algorithms. (**c**) Various performance comparisons of the algorithms.

**Figure 11 entropy-24-01678-f011:**
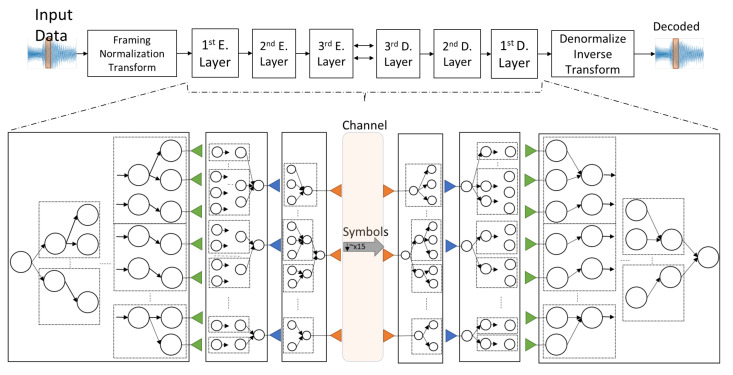
Multi-level encoder and decoder cortical coding architecture. See text for explanation.

## Data Availability

The data generation algorithms source code and the data used in the train & tests are publicly available in *DataGeneration* folder at repository [50]. Source code of the Cortical Coding Network Training Algorithm is not publicly available yet. Compiled version is publicly available at repository [51]. All comparison codes used in this work are publicly available in *ComparedClustering* folder at repository [50].

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
