# Peer review of "Brain Inspired Cortical Coding Method for Fast Clustering and Codebook Generation"

_entropy, 2022, doi:10.3390/e24111678_

Round 1

Reviewer 1 Report

This study introduced a feature extraction method inspired by energy-entropy relations of sensory cortical networks in the brain. Overview the paper, this study has good organization, and the issue is very interesting. Some sincerely comments as follows, for the author(s), may be useful to improve your work:

Comment 1. The literature review is very important. Please provide the recent related literatures, and the summary table should be provided. Moreover, please point out the research’s gap. What’s differ with previous review paper?

Comment 2. This comparison of experiment should examined recent clustering methods. Theses method could obtain superior performance than traditional clustering methods such as K-means.    

Comment 3. The sensitive analysis should be provided. The cross validation should be examined in the experiments. The discussion should deeply investigate the theorem implications and related development.

Reviewer 2 Report

In this work, the authors introduced a feature extraction algorithm that is based on energy-entropy relations of cortical networks in the brain. This algorithm demonstrated significant improvement in computational efficiency (no major drop in distortion encoding performance) when compared with traditional algos on a range of simulated datasets. Overall, this manuscript is well-written, all claims are well-supported by the numerical experiment results, and sufficient implementation details are provided. 

I think this work is pretty good in terms of interest to the community, and the soundness of the method as indicated earlier.

The only thing that could be improved is the benchmark (k-means) used in this work. K-means has been there for a while and there are other clustering algorithms that can have better performance than k-means. For example, Expectation–Maximization clustering using Gaussian Mixture Models (GMM), which is more flexible than k-means. K-mean only uses the mean value of the clusters and ignores other useful information, such as the variance. To further improve the quality of the manuscript, consider running more recent/advanced clustering algorithms for the comparison, and report any performance or runtime differences between the proposed algorithm and the newer algorithms (e.g., EM-based clustering using GMM).

Reviewer 3 Report

The article «Brain Inspired Cortical Coding Method for Fast Clustering and Codebook Generation» develops artificial intelligence methods in informatics and biology. The authors suggest new clustering algorithms and compare it with known ones obtaining some gains. This approach is new compared to classical neural networks.

However, there are some comments on the materials and presentation of the article:

1) In the introductory part, when considering vectorization approaches, it is interesting to consider works on matrix data transformations, such as, for example, PCA (https://www2.humusoft.cz/www/papers/tcp05/mudrova.pdf) and SVD (https: //doi.org/10.3390/app11115235)

2) Speaking of execution time, one simply cannot help noticing specialized inference acceleration tools such as OpenVINO

3) Figure 1. The title is too long. Put it in the text with a description, shorten the title.

4) Figure 2. The title is too long. Put it in the text with a description, shorten the title.

5) Figure 3. The title is too long. Put it in the text with a description, shorten the title.

6) Figure 4. The title is too long. Put it in the text with a description, shorten the title.

7) Figure 4 shows a large gain in execution time. This needs to be considered in more detail in the text.

8) It is necessary to indicate the characteristics of the calculator in the introductory part, since then a time comparison takes place.

9) Figure 5. The title is too long. Put it in the text with a description, shorten the title.

10) Figure 5b is better presented as a table, since the lines overlap a lot.

11) Figure 6. The title is too long. Put it in the text with a description, shorten the title.

12) Section 7 Methods. In my opinion, it should be taken out in the middle. According to the classical structure, the reader expects that the last sections of the article will be "Results", "Discussion" and "Conclusion". It should not come after "Conclusion", in the view of the reviewer.

13) Figure M1. The title is too long. Put it in the text with a description, shorten the title.

14) Figure M2. The title is too long. Put it in the text with a description, shorten the title.

15) Algorithm 1. It is better to insert in the text next to the first mention, as is done with Algorithm 2.

16) Figure M4. The title is too long. Put it in the text with a description, shorten the title.

In general, these remarks do not reduce the impression of the article, however, in the opinion of the reviewer, it is considered especially important to supplement the introductory part with references to modern works.

Round 2

Reviewer 1 Report

I have no further comments.